# Esophageal and Head and Neck Cancer Patients Attending Ocean Road Cancer Institute in Tanzania from 2019 to 2021: An Observational Study

**DOI:** 10.3390/ijerph20043305

**Published:** 2023-02-13

**Authors:** Luco P. Mwelange, Simon H. D. Mamuya, Julius Mwaiselage, Magne Bråtveit, Bente E. Moen

**Affiliations:** 1Department of Environmental and Occupational Health, School of Public Health and Social Sciences, Muhimbili University of Health and Allied Sciences, Dar es Salaam P.O. Box 65001, Tanzania; 2Ocean Road Cancer Institute, Dar es Salaam P.O. Box 3592, Tanzania; 3Department of Global Public Health and Primary Care, Centre for International Health, University of Bergen, 5020 Bergen, Norway

**Keywords:** occupation, head and neck cancers, esophageal cancer, Tanzania

## Abstract

Background: Cancer in Africa is an emerging public health problem that needs urgent preventive measures, particularly in workplaces where exposure to carcinogens may occur. In Tanzania, the incidence rate of cancer and mortality rates due to cancers are increasing, with approximately 50,000 new cases each year. This is estimated to double by 2030. Methods: Our hospital-based cross-sectional study describes the characteristics of newly diagnosed patients with head and neck or esophageal cancer from the Ocean Road Cancer Institute (ORCI), Tanzania. We used an ORCI electronic system to extract secondary data for these patients. Results: According to the cancer registration, there were 611 head and neck and 975 esophageal cancers recorded in 2019–2021. Two-thirds of these cancer patients were male. About 25% of the cancer patients used tobacco and alcohol, and over 50% were involved in agriculture. Conclusion: Descriptions of 1586 head and neck cancer patients and esophageal cancer patients enrolled in a cancer hospital in Tanzania are given. The information may be important for designing future studies of these cancers and may be of value in the development of cancer prevention measures.

## 1. Introduction

Cancer incidence and mortality are sharply increasing worldwide [1]. Cancer ranks first or second in causing deaths in most countries [1,2]. The increase in cancer cases reflects the change and distribution of risk factors, including environmental exposures, in different regions. It is projected that the number of new cancer cases in Africa will nearly double by 2030 [3]. Clearly, cancer in Africa is an emerging public health problem that needs urgent preventive measures and more knowledge [3,4]. About 50,000 new cancer cases are reported annually in Tanzania, and it is predicted that by 2030, this figure will have doubled [5].

Many factors influence the development of cancer, including genetic, environmental, and occupational factors. Exposure to the latter two factors is particularly important in many countries. In Tanzania, many workers are employed in agriculture, where many may be exposed to carcinogenic pesticides. In addition, industrial workplaces are increasing in number and with the accompanying exposure to carcinogenic agents.

Currently, the prevalence of occupational cancers is unknown in Tanzania, and studies are warranted for the risk evaluation and implementation of preventive measures against cancer. Head and neck and esophageal cancers are common cancer types in Tanzania, and evidence from developed countries has shown that these could be related to occupational exposures.

Head and neck cancers are the sixth most common cancer type worldwide, with an incidence per year of 550,000–690,000 and an annual mortality of 300,000–450,000 [6,7,8,9,10]. Almost 67% of the new cases and 82% of the mortality due to head and neck cancers occur in developing countries [11,12]. Hospital data from Tanzania show that head and neck contributes to 7% of all cancer cases [5]. Head and neck cancers might be associated with a range of risk factors, such as alcohol use, tobacco use, poor oral health, dietary deficiencies, and human papillomavirus (HPV) [7,13,14]. Several occupational exposures have also been suggested as risk factors for different head and neck cancers [15]; for instance, diesel exhaust, asbestos, organic solvents, metal dust, asphalt, wood dust, stone dust, mineral wool, cement dust, chlorinated, oxygenated, and petroleum solvents [16,17]. Asbestos and strong acids, for example, have been linked to larynx and laryngeal cancer [18]. Hexavalent chromium, leather, and wood dust are linked to nasal cancer [15,19,20,21]. Occupational exposure to perchloroethylene or trichloroethylene increases the risk of head and neck cancers in women [17].

Esophageal cancer is the seventh most common cancer globally, with more than 500,000 cases annually contributing to over 500,000 cancer deaths per year [22]. Esophageal cancer incidence and mortality are increasing in developing countries [23,24]. On the African continent, both the South and East have high numbers of cases and mortality [25]. East Africa has unique features with high numbers of such cases occurring at a young age [23]. Reported risk factors for esophageal cancer include tobacco use, alcohol use, low socio-economic status, hot food, and pickled food [23,26,27,28]. Occupational exposure to dust and metal exposure among workers has shown a significant increased risk of esophageal cancer [29]. Additionally, it has been proposed that exposure to polycyclic aromatic hydrocarbons (PAHs) is a risk factor for developing esophageal cancer [30]. Other common chemicals such as asbestos, hydrocarbons and their derivatives, chlorinated solvents, and pesticides might be associated with esophageal cancer [31,32]. 

Occupational exposure differs between men and women, and evidence suggests that gender differences in occupational exposure are partly explained by the segregation of the workforce [33]. Men and women have markedly different occupational exposure patterns, even within the same occupation [34]. 

Most of the evidence relating to risk factors for esophageal and head and neck cancers comes from developed countries, and there is little evidence from developing countries [35]. However, there is growing evidence linking esophageal and head and neck cancers to environmental exposures [22]. We, therefore, wanted to describe possible risk factors for head and neck and esophageal cancer in Tanzania, such as age, gender, family cancer, residency, occupation, smoking, and alcohol use. We decided to perform this study at a cancer hospital in Tanzania, where this information was available.

The results of this study might be of importance in identifying factors that can be used in future studies of risk factors for cancer, as well as for preventive strategies for cancer in Tanzania. 

## 2. Materials and Methods

### 2.1. Study Setting

This study was conducted at Ocean Road Cancer Institute (ORCI), Dar es Salaam, which is a public national specialized hospital for cancer treatment in Tanzania. ORCI was established in June 1996 by the Ocean Road Cancer Institute Act No. 192 of 1996. It is the only hospital in Tanzania that offers all types of cancer services, including screening, radiotherapy, chemotherapy, and palliative care. It is the national referral hospital for cancer and serves patients from all over the country, as well as other nearby countries. 

Tanzania has a population of 61 million people according to the 2022 census [36]. Over 62% of its population is engaged in the agriculture sector, 32% work in the services sector, and 8% work in manufacturing industries [37].

### 2.2. Study Design and Population 

This hospital-based descriptive cross-sectional study comprises all esophageal and head and neck cancer patients diagnosed or referred to ORCI from 2019 to 2021. During the observation period, there were 3839, 3988, and 4026 new cancer patients in this hospital in the years 2019, 2020, and 2021, respectively.

The study included 611 cases of head and neck cancer and 975 cases of esophageal cancer. We extracted information about these cases from the ORCI electronic system. The hospital system used in this study has 3 different sections for recording patient information and data. The first section records general information such as name, identification number, date of birth, sex, and last visit. Another section records the patient’s address and includes the country, region, district, ward, and village/street where the patient lives. The last section recorded the patient’s medical history. Information in all these sections was filled in by the physicians.

### 2.3. Data Sources, Collection, and Procedures 

The secondary data obtained from the hospital contained a specific patient identification number used only for data verification. Data such as socio-demographics, referrals, and treatment-related information were extracted from the system and stored in an Excel file. Before data extraction, we used one day to train two research assistants on the extraction procedures. The project investigator ensured day-to-day quality control of the data collected. We extracted the following information for each patient: age, sex, region and district of residence, year of diagnosis, occupation, type of cancer (esophageal and head and neck cancers), smoking and alcohol use, family history of cancer, type of cancer, and the family relationship.

### 2.4. Data Management and Statistical Analysis

We checked for consistency and the harmonization of collected data. Occupations were coded according to the main groups of the International Standard Classification of Occupations (ISCO 08). In these data, there was some missing information in the family history of cancer (31%), alcohol use and smoking (20%), and occupational status (2%), and over 98% of occupation-recorded data, 11% were not correctly registered. Any missing information about the patients was coded as unknown. Patients recorded as children, dependents, or retired were coded as unclassified. Family relationships were grouped into first, second, and third degree. First-degree relationships included parents and children; second-degree included brothers, sisters, grandparents, and grandchildren; and third-degree included uncles, aunts, great-grandparents, and great-grandchildren.

The following categorical variables were used: age groups (<40, 40–60, and >60 years), family history of cancer (yes/no), family relationship degree (1–3), occupation (managers, professionals, technicians, and associate professionals, services and sales workers, agricultural workers, craft and related trades workers, plant and machine operators, and assemblers, unclassified and unknown), tobacco use (yes/no), and alcohol use (yes/no). Categorical variables were described by frequency and percentage.

The regions where the patients came from were used to indicate the spatial distribution of cancer cases across the country. To calculate incidence rates, we added the cancers for the period of three years for each region by using the National Bureau of Statistics of Tanzania [36], which provided population projection for the three years for each region. Then, we divided the total number of cancer cases for the period of three years by the total population for the three years. The ratio obtained was multiplied by 100,000 person-years. The value obtained (cancer incidence per person-years) was used to compare cancer cases across regions in Tanzania using a spatial distribution map.

Because both tobacco and alcohol use increase the risk of developing cancer, we decided to make a variable called ‘tobacco and alcohol use’, computed from those cancer patients who both smoked and drank alcohol. 

The statistical package SPSS version 23 was used for data analysis. The continuous variables were presented with means and standard deviations. Chi-squared tests were used to assess the relationship between categorical variables. A *p*-value less than 0.05 was considered statistically significant.

Ethical approval was obtained from the Senate Research and Publications Committee of Muhimbili University of Health and Allied Sciences (I.R.B. #: MUHAS-REC-12-2021-914). The Director General of the ORCI gave us permission to use the data. We ensured confidentiality during the data extraction and analysis. The names of the cancer patients were not included in the database of secondary data.

## 3. Results

### 3.1. Socio-Demographic Characteristics

A total of 1586 new head and neck and esophageal cancers cancer cases were registered in ORCI during the three-year study period (2019–2021). Out of these, 611 (38.5%) were head and neck cancer patients, with a mean age of 53 (SD = 16) years for males and 58 (SD = 14) for females; according to the Tanzania Bureau of Statistics in 2020, the expectancy of Tanzania was 64 to 69 years for males and females, respectively. In all, 56% of male and 72% of female patients were below 60 years of age. About 50% of male and 24% of female patients used alcohol; 43% and 6% of male and female patients used tobacco. The results also showed that less than 10% of cancer patients had a family history of cancer (Table 1).

The number of esophageal cancer patients was 975, which is 61.5% of the 1586 cases registered, and they had a mean age of 58 years for males and 59 years for females. More than 53% of male and 57% of female patients were under 60 years of age. Over 64% of esophageal cancer patients were male. About 50% of male and 12% of female patients used alcohol, and about 43% of males and 12% of females used tobacco. More than 37% of the males both used alcohol and smoked. For females, 12% smoked and used alcohol. Over 98% of the patients had their occupational status recorded in the system, and about 50% were involved in the agriculture sector for esophageal and head and neck cancers. We performed a chi-squared test of independence to examine the relationship between gender and characteristics of interest for both cancers, head and neck and esophageal. For head and neck cancer, the relation between characteristics, age group, cancer family history, alcohol use, tobacco use, and occupation between male and female was significant with *p* < 0.01. However, for esophageal cancer, only alcohol use, tobacco use, and occupation were significant with *p* < 0.01. 

Further examination of head and neck cancer patients revealed that the most common types of head and neck cancer were laryngeal (35%) and nasopharyngeal (22%) cancers. The results also showed that smoking and alcohol were most common among laryngeal cancer patients (Table 2).

### 3.2. Geographical Distribution

Figure 1 classifies all the administrative regions of Tanzania according to the incidence rate per 100,000 person-years. At the time, Kilimanjaro and Dar es Salaam had the highest incidence rates of head and neck cancers, with 0.970 and 1.07 cases per 100,000, respectively. Dar es Salaam and Kilimanjaro had the highest incidence rates for esophageal cancer, with 1.52 and 1.90 cases per 100,000 person-years, respectively. Generally, the south-eastern and northern parts of Tanzania had higher incidence rates than other parts. 

## 4. Discussion

To our knowledge, this is the first study in Tanzania to describe and compare the characteristics of head and neck and esophageal cancer patients considering any possible occupational aspects. In total, 611 head and neck and 975 esophageal cancers were registered at ORCI in 2019–2021. About half of the patients were agricultural workers, 53% for esophageal and 47% for head and neck cancer. Two-thirds of these cancer patients were male. A higher proportion of males smoked and used alcohol than females for both head and neck and esophageal cancer patients. 

In the present study, over half of the head and neck cancer patients were engaged in agriculture. This result is consistent with recent research from Nigeria, where it was shown that more than 36% of the patients worked in agriculture [38]. In the case of esophageal cancer patients, the current study shows that almost 53% are engaged in agriculture. More than 40% of this group’s patients were engaged in agriculture in Ethiopia and Tanzania, where related research was carried out [23,27]. These findings contrast with those in developed countries where cancer patients engaged in agriculture ranged from 1.1 to 2.4% [39,40] for both head and neck and esophageal cancer patients. These findings may reflect that most of the population in developing countries, such as Tanzania, are engaged in agriculture. 

Occupational exposure to agriculture has been studied earlier as a risk factor for cancers, including both head and neck and esophageal cancer. A case-control study in Iran found that those engaged in agriculture had an odds ratio of 3.26 95% CI (1.13–9.43) of developing head and neck cancer compared to non-agriculture patients [41]. A possible risk factor in agriculture might be exposure to pesticides. For example, a review paper revealed an association between pesticide exposure and head and neck cancers [42]. On the other hand, this finding contradicts another review study suggesting that head and neck cancers are not associated with pesticide exposure [43]. More research is needed in this area.

A multicenter cross-sectional study conducted in Turkey showed that pesticide exposure might give an increased risk of esophageal cancer, but the findings were not conclusive [44]. These studies were conducted where exposures might be low compared to our context, where unsafe handling prior to, during, and after the use of pesticides may cause higher exposures [45]. 

In the present study, we found that head and neck cancers are more common among men (66%) [46]. These results match earlier studies conducted in low- and middle-income countries. For example, a cross-sectional study of head and neck cancers in India, using secondary data from a hospital, showed that the proportion of males ranged from 68% to 70% [12,47]. Another cross-sectional study of head and neck cancers in South Africa using secondary data from the hospital showed that 71% were male [13]. We also observed a higher proportion of male patients with esophageal cancer (64%). This finding was similar to a previous case-control study of esophageal cancer in Tanzania, where the proportion of males was 62% [23]. Moreover, this finding was similar to a case-control study conducted in Kenya, where the proportion of males was 66% [48]. This might be caused by different exposure to environmental factors at work and differences in personal factors, such as smoking or not, when comparing males and females. In our study, the males smoked considerably more than the females.

A unique feature of esophageal and head and neck cancers in developing countries is that it is common among relatively young people [23]. The present study found a similar finding with a mean age of 55 years for head and neck cancers. This finding is similar to other cross-sectional studies conducted in Benin using secondary data, which had a mean age of 58 years [14]. Moreover, this study is consistent with another cross-sectional study using secondary data conducted in Ethiopia, which had a mean age of 51 years [27].

This study found that esophageal cancer patients had a mean age of 58 years, which is almost eight years lower compared to the life expectancy of a Tanzanian person. This result is in line with other case-control studies conducted in Ethiopia, which had a mean age of 55 years [20], and another case-control study conducted in Kenya, which had a mean age of 59 years [48]. The present study found that more than 50% of the study patients for head and neck and esophageal cancers were below 60 years. This finding is similar to other studies conducted in other developing countries, such as Nigeria, where 65% of the study population was below 60 years [38,49]; a study conducted in South Africa, in which over 59% of the study population was below 60 years [13]; and another study conducted in Benin, where almost 60% of the study population was below 60 years [14]. 

In this study, the prevalence of alcohol use among head and neck cancer patients was 39%. This finding was lower than in other studies in lower- and middle-income countries. A cross-sectional study using secondary data conducted in South Africa found that 71% used alcohol [13]. Additionally, a cross-sectional study conducted in Nepal and India using secondary data reported that alcohol use was over 59% [12,47]. The current study also shows that 27% of the use of alcohol among esophageal cancer patients is lower than other studies conducted in other countries, such as Kenya (39%) [50] and Zambia (48%) [51]. The differences between the countries might be due to differences in culture regarding alcohol consumption. Other possible explanations for these differences in results might be due to differences in study design, participant recruitment, and methods of obtaining the information.

This study showed that tobacco use among head and neck cancer patients was 25%, which is higher compared to the smoking prevalence among the general population in Tanzania (8.7%) [52]. However, this result differs from other findings from the cancer case studies conducted in lower- and middle-income countries, such as South Africa, where the smoking prevalence was 86% [17]; and India [53], where it was over 45%. However, the smoking prevalence in the general population also differs between countries as well: India (30%) [53], South Africa (22%) [54], and Tanzania (8%) [52]. 

In this study, Tanzania’s eastern and northern parts had higher cancer incidence rates than other parts. This finding is similar to another study conducted in Tanzania [55], which also found that Tanzania’s eastern and northern regions showed higher incidence rates of esophageal cancer. The cause for this finding is not clear, and further studies are needed to clarify. The higher prevalence of the disease in eastern Tanzania may be connected to greater accessibility to care. ORCI’s location in the eastern part may be the reason for this similarity. It is important to note that some of the lowest incidence regions are not considerably farther apart from some of the higher incidence ones. For instance, Kilimanjaro, a location in the highest incidence category, is not considered further away from ORCI than Morogoro, Iringa, and Manyara, which are regions in the lowest incidence group. We have no information about the reason why we find these geographical differences. Both genetic and environmental factors might be of importance. This topic should be in focus in future studies in Tanzania.

Oropharyngeal cancer was diagnosed among 8.8% of the cancer patients in this study. Human papilloma virus (HPV) is known to be associated with this cancer type [56]. Unfortunately, we had no information about the HPV status of the patients in this study. In Tanzania, the HPV vaccination of 14-year-old girls was introduced nationally in April 2018. Our study period was 2019–2021. 

To our knowledge, this is the first study in Tanzania to compare descriptive characteristics on head and neck and esophageal cancer. However, this study has some limitations. The registrations were made before the present study started. This may have caused missing and improper registration of information about the patients in the hospital records. However, the information used for the study was written in the system by the physicians, making it likely that the data quality is higher than if technical personnel provided the data. Several variables provided us with too few details; for instance, we do not know if tobacco smoking was previous or ongoing. Therefore, this reduces the possibility of making any causal conclusions. The study design is weak, as it only describes the patients and does not compare with any other groups of patients or populations. For future studies of these types of cancer, case-control studies might be useful if causative agents are explored. The choice of controls will be a challenge for such studies. Cohort studies are another design that might be used for these types of research questions, but they require more resources. The inclusion of controls will also be important in cohort studies. 

In Tanzania, there are few hospitals that specifically diagnose and treat cancer patients. Using data from the ORCI was the optimal method in the present study. This might change over time, and in the future, it might be possible to include other hospitals and health facilities. Another possibility for future studies is to establish cooperation with similar cancer hospitals in other East African countries. 

The results from the present study might not be generalized to other countries, because they are based on data from a selected group of cancer patients in Tanzania. The results should be interpreted with caution. However, the findings might be of interest to health personnel working in cancer in Tanzania and similar countries and may inspire further studies of the topic.

## 5. Conclusions

In 2019–2021, 611 head and neck cancer patients and 975 patients with esophageal cancer were enrolled in a cancer hospital in Tanzania. About two thirds of the patients were men. Alcohol and smoking were commonly used by these patients, more among men than among women. Half of the patients were working in agriculture, and the number of cases was unevenly distributed in the country. The results might be useful for the development of cancer prevention measures, as well as for future studies on cancer. Further studies are recommended.

## Figures and Tables

**Figure 1 ijerph-20-03305-f001:**
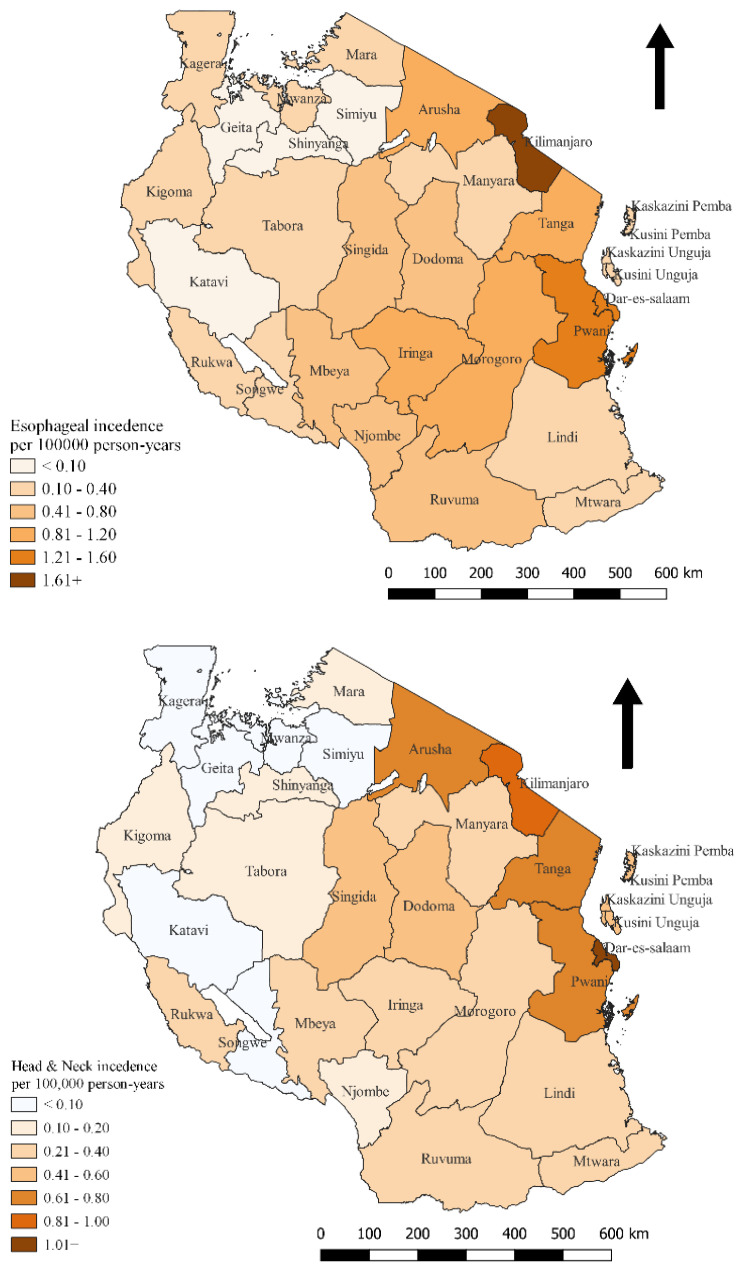
The figures show the number of cancer incidences per 100,000 person-years.

**Table 1 ijerph-20-03305-t001:** Socio-demographic characteristics among head and neck and esophageal cancers patients at the cancer hospital in Tanzania 2019–2021. Men and women are compared using chi-square tests.

Characteristics	Head & Neck Cancers		*p*-Value	Esophageal Cancer		*p*-Value
	Male n (%)	Female n (%)		Male n (%)	Female n (%)	
**Total**	406 (66.4)	205 (33.6)		621 (63.7)	354 (36.3)	
Age; mean (SD)	53 (16)	58 (14)		58 (14)	58 (14)	
**Age group**						
<40	72 (17.7)	53 (25.9)	<0.01	77 (12.4)	38 (10.7)	<0.32
40–60	166 (40.9)	102 (49.7)		268 (43.2)	170 (48.1)	
>60	168 (41.4)	50 (24.4)		276 (44.4)	146 (41.2)	
**Cancer Family history**						
Yes	18 (4.4)	8 (3.9)	<0.01	47 (7.6)	28 (7.9)	<0.67
No	274 (67.5)	122 (59.5)		395 (63.6)	215 (60.7)	
Unknown	114 (28.1)	75 (36.6)		179 (28.8)	111 (31.4)	
**Relationship degree ^a^**						
First	6 (33.3)	3 (37.5)	<0.10	29 (61.7)	15 (53.6)	<0.92
Second	12 (66.7)	3 (37.5)		16 (34.0)	11 (39.3)	
Third	0 (0.0)	2 (25.0)		2 (4.3)	2 (7.1)	
Alcohol Use						
Drinkers	217 (53.5)	50 (24.4)	<0.01	316 (50.9)	93 (26.3)	<0.01
Never drinker	126 (31.0)	112 (54.6)		181 (29.1)	178 (50.3)	
Unknown	63 (15.5)	43 (21.0)		124 (20.0)	83 (23.4)	
Tobacco use						
Smokers	178 (43.8)	13 (6.3)	<0.01	263 (42.4)	43 (12.1)	<0.01
Never smoker	167 (41.2)	150 (73.2)		232 (37.4)	228 (64.4)	
Unknown	61 (15.0)	42 (20.5)		126 (20.2)	83 (23.4)	
Smoking + alcohol	149 (36.7)	11 (5.4)		223 (35.9)	33 (9.3)	
**Occupation (ISCO 08 code)**						
Managers	36 (8.9)	22 (10.7)	<0.01	46 (7.4)	27 (7.6)	<0.01
Professionals	26 (6.4)	12 (5.9)		34 (5.5)	13 (3.7)	
Technical and associate Professionals	10 (2.5)	2 (0.9)		30 (4.8)	5 (1.4)	
Services and Sales workers	35 (8.6)	38 (18.5)		39 (6.3)	63 (17.8)	
Craft and related Trade workers	15 (3.7)	9 (4.4)		20 (3.2)	6 (1.7)	
Plant and Machine Operators and Assemblers	33 (8.1)	0 (0.0)		39 (6.3)	2 (0.6)	
Agricultural workers	182 (44.8)	106 (51.7)		321 (51.7)	197 (55.6)	
Unclassified	62 (15.3)	12 (5.9)		80 (12.9)	34 (9.6)	
Unknown	7 (1.7)	4 (2.0)		12 (1.9)	7 (2.0)	

^a^ Among those with family cancer.

**Table 2 ijerph-20-03305-t002:** Socio-demographic characteristics among head and neck cancer patients at the Tanzania cancer hospital 2019–2021. ^a^ Among those with family cancer.

Characteristics	Hypo-Pharyngeal Cancer	Laryngeal Cancer	Nasal Cavity Cancer	Naso-Pharyngeal Cancer	Oral Cancer	Oro-Pharyngeal Cancer	Salivary Gland Cancer
	n (%)	n (%)	n (%)	n (%)	n (%)	n (%)	n (%)
**Total**	80 (13.1)	216 (35.4)	31 (5.7)	135 (22.1)	42 (6.9)	54 (8.8)	53 (8.7)
**Age; mean (SD)**	53 (16)	56 (16)	46 (19)	50 (17)	59 (13)	56 (17)	50 (15)
Sex							
Male	42 (52.5)	168 (77.8)	18 (58.1)	92 (68.1)	25 (59.5)	34 (63.0)	27 (50.9)
Female	38 (47.5)	48 (22.2)	13 (41.9)	43 (31.9)	17 (40.5)	20 (37.0)	26 (49.1)
**Age group**							
<40	18 (22.5)	28 (13.0)	13 (41.9)	40 (29.6)	4 (9.5)	8 (14.8)	14 (26.4)
40–60	34 (42.5)	99 (45.8)	11 (35.5)	56 (41.5)	18 (42.9)	22 (40.7)	28 (52.8)
>60	28 (35.0)	89 (41.2)	7 (22.6)	39 (28.9)	20 (47.6)	24 (44.5)	11 (20.8)
**Cancer Family history**							
Yes	3 (3.7)	12 (5.6)	2 (6.5)	3 (2.2)	2 (4.8)	3 (5.6)	1 (1.9)
No	51 (63.8)	142 (65.7)	24 (77.4)	83 (61.5)	29 (69.0)	35 (64.8)	32 (60.4)
Unknown	26 (32.5)	62 (28.7)	5 (16.1)	49 (36.3)	11 (26.2)	16 (29.6)	20 (37.7)
**Relationship degree ^a^**							
First	1 (33.3)	2 (16.7)	0 (0.0)	2 (66.7)	2 (100.0)	2 (66.7)	0 (0.0)
Second	2 (66.7)	9 (75.0)	2 (100.0)	1 (33.3)	0 (0.0)	0 (0.0)	1 (100.0)
Third	0 (0.0)	1 (8.3)	0 (0.0)	0 (0.0)	0 (0.0)	1 (33.3)	0 (0.0)
**Alcohol Use**							
Drinkers	31 (38.7)	118 (54.6)	10 (32.3)	54 (40.0)	22 (52.4)	18 (33.3)	14 (26.4)
Never drinker	38 (47.5)	63 (29.2)	17 (54.8)	53 (39.3)	16 (38.1)	23 (42.6)	28 (52.8)
Unknown	11 (13.8)	35 (16.2)	4 (12.9)	28 (20.7)	4 (9.5)	13 (24.1)	11 (20.8)
**Tobacco use**							
Smokers	27 (33.8)	92 (42.6)	2 (6.5)	33 (24.4)	12 (28.6)	16 (29.6)	9 (17.0)
Never smoker	42 (52.5)	90 (41.7)	25 (80.6)	76 (56.3)	26 (61.9)	25 (46.3)	33 (62.2)
Unknown	11 (13.7)	34 (15.7)	4 (12.9)	26 (19.3)	4 (9.5)	13 (24.1)	11 (20.8)
Smoking + alcohol	24 (30.0)	75 (34.7)	1 (3.2)	31 (23.0)	10 (23.8)	13 (24.1)	6 (11.3)
**Occupation (ISCO 08 code)**							
Managers	8 (10.0)	22 (10.2)	2 (6.5)	13 (9.6)	3 (7.1)	5 (9.3)	5 (9.4)
Professionals	6 (7.5)	10 (4.6)	4 (12.9)	11 (8.1)	2 (4.8)	1 (1.9)	4 (7.5)
Technical and associate Professionals	2 (2.5)	5 (2.3)	1 (3.2)	1 (0.7)	1 (2.4)	2 (3.7)	0 (0.0)
Services and Sales workers	17 (21.3)	24 (11.1)	4 (12.9)	14 (10.4)	4 (9.5)	4 (7.4)	6 (11.3)
Craft and related Trade workers	2 (2.5)	11 (5.1)	0 (0.0)	1 (0.7)	4 (9.5)	2 (3.7)	4 (7.5)
Plant and Machine Operators and Assemblers	1 (1.3)	17 (7.9)	1 (3.2)	8 (5.9)	2 (4.8)	2 (3.7)	2 (3.8)
Agricultural workers	37 (46.3)	98 (45.4)	14 (45.2)	68 (50.4)	18 (42.9)	29 (53.7)	24 (45.3)
Unclassified	6 (7.5)	26 (12.0)	4 (12.9)	16 (11.9)	8 (19.0)	7 (13.0)	7 (13.2)
Unknown	1 (1.3)	3 (1.4)	1 (3.2)	3 (2.2)	0 (0.0)	2 (3.7)	1 (1.9)

## Data Availability

Data for this study that supports its findings are available upon reasonable request from the corresponding author.

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
