# Peer review of "Esophageal and Head and Neck Cancer Patients Attending Ocean Road Cancer Institute in Tanzania from 2019 to 2021: An Observational Study"

_ijerph, 2023, doi:10.3390/ijerph20043305_

Round 1

Reviewer 1 Report

I have reviewed this interesting epidemiological study about esophageal (n = 975) and head and neck cancer (n = 511) in a institute in Tanzania that is the national referral hospital for cancer.

I have appreciated so much the effort of the authors in performing this institutional study considering the lack of a national cancer database.

Abstract: the topics of the structured Abstract should be included (Introduction, aim, Methods and so on).

Introduction: The authors show a special concern with occupational causes of the studied cancers, however, in their aims, the looked for association between cancer and risk factors as occupation, smoking and alcohol use without a specific survey about the several occupational exposures mentioned.

Methods: Since the study is retrospective, I wonder if some data could have been missed for such an epidemiological study. I think this paper fails in demonstrate specific findings on the relation between occupational exposure and cancer.

I think it is a problem to gather all the different sites of head and neck cancer in a sole analysis. Were thyroid, salivary glands and nasopharyngeal cancers mixed?

Results: Table 1: Hard can I understand what exactly the authors try to compare statistically, since this is predominantly an epidemiological study.

Figure 1 shows absolute number of cases. In the absence of populational data, it is not possible to assure that there was some variation in the incidence.

Figure 2: I ask the authors to clarify in the Methods how the interesting data of this figure were extracted.

The Conclusion should be more objective and not based in inferences.

Reviewer 2 Report

- In results section, please add discussion of average life expectancy of a person in Tanzania. This will help put the mean ages into context. 

- Page 6, line 211 change "non-cancer" to "non-agriculture"

- A discussion of HPV vaccination rates would be helpful. Any info on HPV positivity of these tumors, or on vaccination rates/patterns? If missing this information, please describe why.

- Page 7 lines 240-248 : again, please put into context of average life expectancy.

-P. 7, lines 273-274: any theories re: Kilimanjaro and the high incidences?

Round 2

Reviewer 1 Report

I have reviewed once again this interesting epidemiological study about esophageal and head and neck cancer in an institute in a national referral hospital for cancer. The paper improved after the corrections performed. However, some aspects are hard to change: some data could have been missed for such an epidemiological study; and different primary sites were gathered in a sole analysis.

Author Response

Our response is in the enclosed file, but can also be read here:

REVIEWER 1 COMMENTS

Manuscript ID: ijerph-2135072 – 2nd round of revision

Thank you for the comments and our opportunity to improve the text. Our response is as follows:

  1. a) We understand the reviewer is not satisfied with the design of our study. This is a descriptive study, and we agree that the design is not optimal. We cannot change the design at this stage of the study, but we have added more text to the methodological discussion of limitations, to show this agreement with the reviewer. We added, line 313-322:

For future studies of these types of cancer, case-control studies might be useful if causative agents are explored. The choice of controls will be a challenge for such studies. Cohort studies is another designed that might be used for this type of research questions but requires more resources. Inclusion of controls will be important also in cohort studies.

In Tanzania, there are few hospitals that specifically diagnose and treat cancer patients. Using data from the ORCI was the optimal method in the present study. This might change over time, and in the future, it might be possible to include other hospitals and health facilities. Another possibility for future studies is to establish cooperation with similar cancer hospital in other East African countries. 

  1. b) Also, the reviewer is not satisfied with our conclusion. We have revised the conclusion, line 22-27 in the abstract to be more in line with the aims and the conclusion at the end of the discussion:

Descriptions of 1586 head and neck cancer patients and esophageal cancer patients enrolled in a cancer hospital in Tanzania are given. The information may be important for designing future studies of these cancers and may be of value in the development of cancer prevention measures.

  1. c) To make the methods clearer, some minor changes have been made in 2.2. Study design and population, lines 102 and 112. We have also revised the language by minor changes in the text.
